# Evolutionary Analysis of the Acoustics of the Baroque Church of San Luis de los Franceses (Seville)

Enedina Alberdi *, Miguel Galindo  and Ángel L. León-Rodríguez

Instituto Universitario de Arquitectura y Ciencias de la Construcción (IUACC), Escuela Técnica Superior de Arquitectura, Universidad de Sevilla. Av. Reina Mercedes 2, 41012 Seville, Spain; mgalindo@us.es (M.G.); leonr@us.es (Á.L.L.-R.)
* Correspondence: ealberdi@us.es

**Abstract:** In the 16th century the Society of Jesus built a large number of churches following the Tridentine model of a Latin cross and a single nave. However, the shift towards this model did not entail the abandonment of the central floor plan, especially in the 17th century. The acoustics of these spaces can present phenomena linked to focalizations which increase the sound pressure level. The church of San Luis de los Franceses, built by the Jesuits for their novitiate in Seville (Spain), is an example of a Baroque church with a central floor plan. Although the church has hosted different congregations since its inauguration it is currently desacralized and used for theatres and concerts. The acoustics of this church were studied by the authors through in situ measurements and virtual models. The main objective was to analyse the evolution and perception of its sound field from the 18th to 21st centuries, considering the different audience distributions and sound sources and the modifications in furniture and coatings. Analysis of the evolution of its sound field shows that the characteristics have remained stable, with a notable influence of the dome on the results for the different configurations studied.

**Keywords:** worship space acoustics; acoustics simulation; acoustic heritage



## 1. Introduction

After the Council of Trent (1545–1563), in his book *Instructiones Fabricae et Supellectilis Ecclesiasticae* [1] Cardinal Carlos Borromeo recorded his "*Instructions for ecclesiastical construction and decoration*", identifying the Latin cross plan with a single nave as the most suitable for churches. This was considered a symbol of Christianity, and was favoured over central floor plans that were more characteristic of pagan temples at the time. The importance of acoustics in churches inevitably leads to the debate on the best way to cover them, as Sendra and Navarro [2] have analysed based on the documentation of four churches (three Jesuit churches and one Franciscan one), identifying the model of single nave churches with wooden roofs as the best option.

The main Jesuit church, Il Gesú (1568–1584) by Giacomo Barozzi da Vignola, follows the Tridentine model and the austere spirit of the Society of Jesus, where the main functions of preaching and administration of the sacraments benefit from a model with a single nave where the parishioners gather. This guarantees an adequate visual connection with the presbytery, while the members of the religious community and schoolchildren of the Society were housed in tribunes which allowed them to follow the services independently from the people. Although this initial model of a Latin cross plan and single nave was reproduced by the Jesuits in numerous churches, circular, Greek cross or elliptical plans were also used to a lesser extent.

The rapid expansion of the Jesuit Order, following its foundation in 1534, allowed the establishment, on 7 January 1554, of the province of Andalusia, with the kingdoms of Jaén, Córdoba, Granada and Seville, as well as the region of Fregenal de la Sierra, south of Badajoz and the Canary Islands [3].

The modo nostro followed by the Jesuits required them to send the designs or plans of the buildings built in each province to Rome for review. This required a degree of coordination and standardization in terms of building typologies. In Andalusia, the churches built in the 16th century mainly follow the model of a single nave, transept, vaults and hemispherical dome [4]. Notable examples of this model are the church of Santa Catalina in Córdoba (1564–1589), the Anunciación in Seville (1565–1579), and the Encarnación in Marchena (Seville) (1584–1593). As a result, of the centralizing tendencies of the Jesuits, throughout the 17th century churches with central typologies were introduced, especially for those cases which required churches with a smaller capacity.

In Andalusia the first example of this model was the church of San Hermenegildo in Seville (1614–1620), with an elliptical plan. This was followed by the church of the College of San Sebastian in Malaga (1626–1630), with a circular plan, churches with the Greek cross floor plan, as seen in the churches of the College of San Torcuato de Guadix (Granada) (1626) and that of the Novitiate of San Luis de los Franceses in Seville (1699–1731). This study focuses on the Baroque church of San Luis de los Franceses, considered a jewel both among the churches of the city of Seville and those built by the Order in the province of Andalusia (Figure 1).

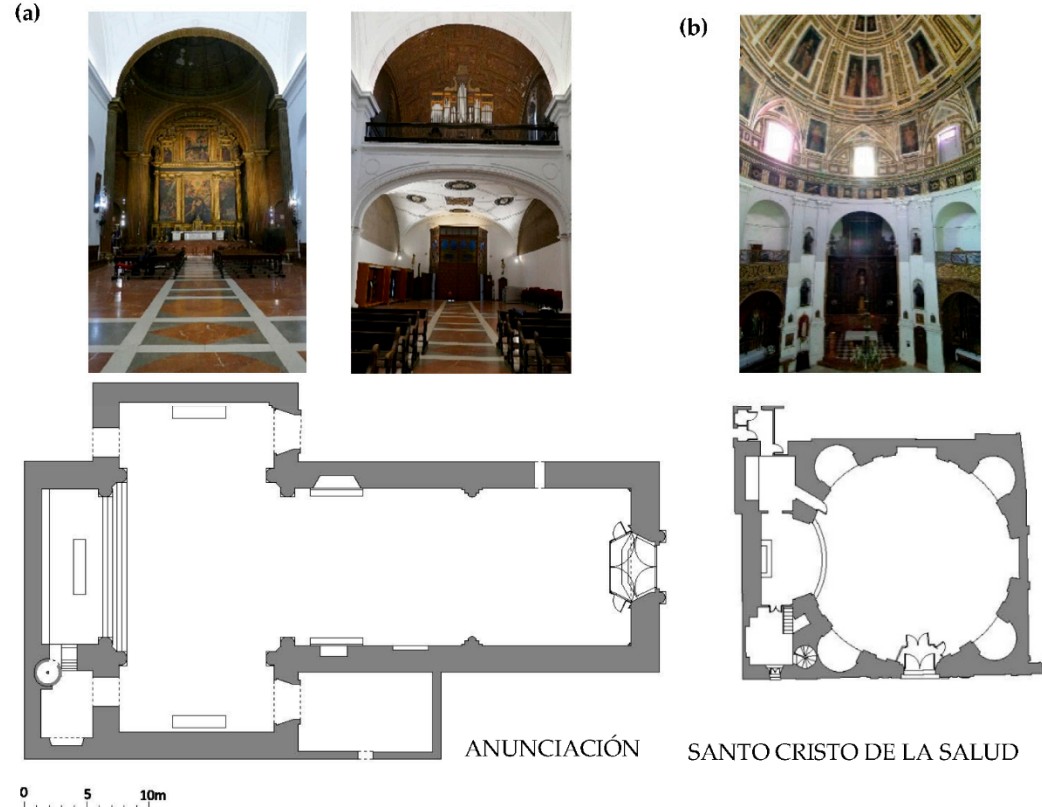

**Figure 1.** Jesuit church. (**a**) Single nave. Church of the Anunciación (Seville). (**b**) Central plan. Church of Santo Cristo de la Salud (Málaga).

Archaeoacoustics, as a method of analysis of historical heritage, makes it possible to study the complex relationship established between architecture and acoustics, introducing as an aspect of analysis the characteristics of the sound field of the analysed spaces, as well as the relationship between people and sound [5,6]. Ecclesial spaces from different historical periods have been analysed from this point of view: Byzantine [7]; Romanesque, such as the Cathedral of Santiago de Compostela [8] or the Abbey of Cluny [9]; Gothic, such as the Spanish cathedrals [10]; and Baroque, such as the Church of Santa María Magdalena [11], as well as unique spaces such as the Mosque Cathedral of Córdoba [12].

The conditions of the sound field in Baroque central spaces have been studied by Cirillo et al. [13] for the churches of St. Luca e Martina (Rome), St. Agnese in Agone (Rome), St. Lorenzo (Turin) and the Basilica of Superga (Turin), where in situ measurements were carried out based on the statement that in all cases "*the small dimension of the church (in plan) allow short source–receiver distance with relatively high C$_{80}$ values*". Furthermore, Carvalho analysed the church of Dos Clérigos in Porto [14], presenting the main monaural parameters derived from impulse responses together with the RASTI index, conducting subjective studies in relation to the evaluation of intelligibility and live music performances.

Spaces with large domes can give rise to unexpected acoustic phenomena [15–17] and focalizations that can increase the sound pressure level or cause colorations in the sound or echoes [18,19]. These aspects have been studied by Alberdi et al. in the church of San Luis de los Franceses [20], where the Bayesian analysis performed showed the presence of double slopes in the energy decay curves, for the different frequencies, especially when the source is located under the dome or near a lateral altar. The double slope phenomenon could be associated with an uneven distribution of sound energy due to acoustic coupling between different sub-volumes, as confirmed by directional intensity maps. The receivers under the dome receive the early reflections mainly from the side walls, while reflections that are more delayed in time do so from the hemispherical surfaces delimiting the volume of the central dome.

The main aim of this research is the evolutionary analysis of the acoustic conditions of the church of San Luis de los Franceses from its inauguration in 1731 to the present day. The church currently displays the same formal configuration inside, but it is deconsecrated and used to stage cultural events such as the Bienal de Flamenco. Over time, the positions of sources and receivers have undergone variations, and this work analyses the impact on the properties of the sound field, in light of the influence that the large central space covered with the large dome could have on the results obtained.

## 2. The Church of San Luis de los Franceses

### 2.1. Origins and Description

The church of San Luis de los Franceses is located on calle San Luis, in the northern part of the historic centre of Seville (Spain). The church is part of a group of buildings, corresponding to the novitiate of San Luis and belonging to the Society of Jesus (Figure 2).

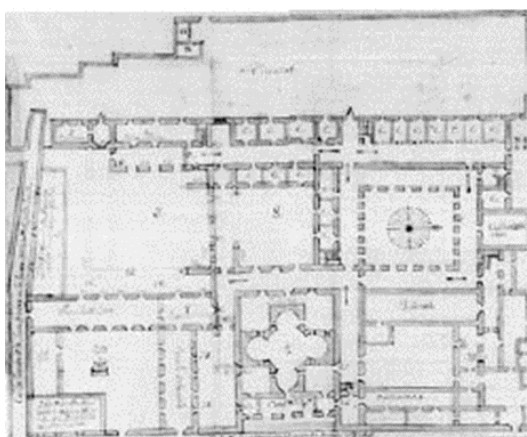

**Figure 2.** Plan of the Royal Convent of San Diego in Seville (1784), formerly known as the Novitiate of San Luis. Archivo Histórico Nacional. Consejos n° 1423.

The Company was established in Seville in 1554. The foundation of the novitiate is thought to date back to 1609, when it was established in some houses near the church of Santa Marina, where there was a small church whose presbytery collapsed in 1695. As a result, work began on the church of San Luis (1699–1731), under the supervision of the architect Leonardo de Figueroa [21,22].

The plan of the church of San Luis follows the model of the Greek cross within a square (Figure 3) so that four semi-circular apses are found at the sides of the cross. The need to separate the interior of the church from calle San Luis led to the incorporation of an atrium before the enclosure, thus preventing direct communication through the apse next to calle San Luis [23].

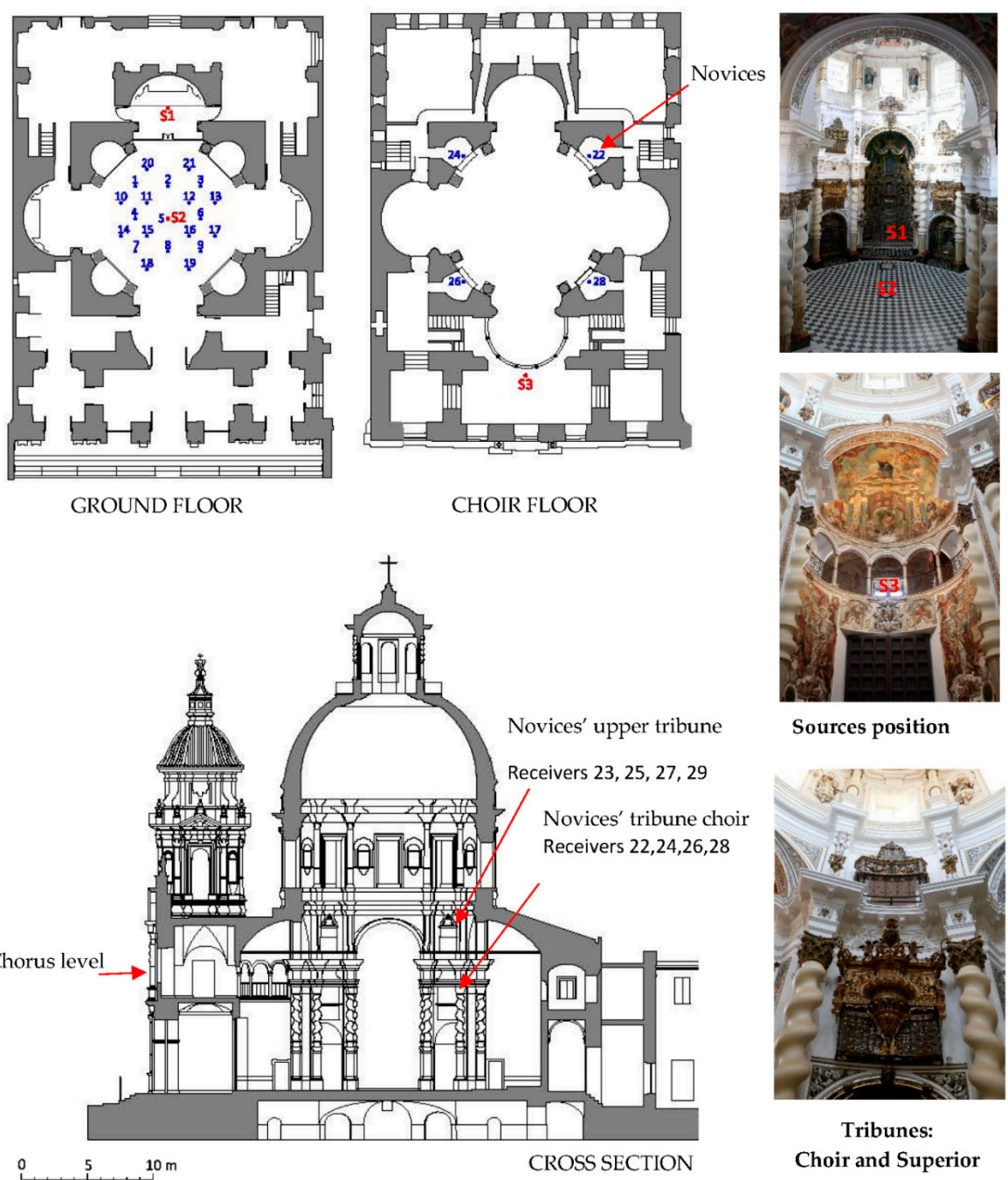

**Figure 3.** Church of San Luis de los Franceses. Ground floor and choir. Cross section.

This space provides enough space on the upper floor for the location of the choir, accessed from the novitiate enclosure on its upper floors. Another aspect to note in the importance of this previous space which provides the interior space, when closed, with better acoustic insulation from the bustle of calle San Luis. The main geometric features of the church are shown in Table 1.

**Table 1.** Main geometric characteristics.

| | |
|---|---|
| Inner volume | 4804 m$^3$ |
| Ground floor surface | 231.80 m$^2$ |
| Choir floor surface | 58.40 m$^2$ |
| Main axis length | 22.35 m |
| Transverse axis length | 22.10 m |
| Inner diameter dome | 12.84 m |
| Higher height under lantern | 34.95 m |

In the layout of the plan, the dome takes on great importance, following the model inspired by Father Pozzo's treatise on perspective [24].

The intervention by Figueroa is clearly seen in the execution of the dome, both in the layout of the drum, with large windows on a cylinder resting on pillars avoiding the use of pendentives, and in the materials used, red brick and ceramic tiles which give the cupola an appearance characteristic of Figueroa's work. The dome rests on four large pillars that connect to small altars on the ground floor. In the same position on the upper floors there are tribunes which provide a visual and acoustic connection between the novitiate and the main space under the dome.

### 2.2. Evolutionary Stages

From its inauguration in 1731 to the present day, the church and its novitiate complex have welcomed different communities and collectives who have used it for different purposes, placing the audience areas, furniture, coatings and sound sources in different positions.

During the 18th and 19th centuries, the church was used by the Jesuits—intermittently, due to their expulsion from Spain in 1767—before the church was ceded to the Franciscan brothers of San Diego in 1784. In the 18th century, the parishioners were not seated on benches or chairs, so for the study hypothesis, the public was considered standing occupying the space under the dome [23]. In addition, the necessary separation between novices and parishioners suggests that the novices occupied the tribunes linking the novitiate with the church, allowing the ceremony to continue with no contact between the two.

During the First Republic (1873), the building continued to be used as a hospice. This meant that during the Spanish Civil War in the 20th century the building was spared and in an acceptable state of conservation, although it remained in disuse and empty from 1968 to 1976, when it opened to worship. It can thus be said that worship was maintained in the church during the 20th century, albeit intermittently, with benches added on the ground floor for the parishioners, following the canons that mark the current liturgy.

Since 1984, major intervention projects have been carried out for the full conservation of the monument. The complex, with its deconsecrated church, served as the headquarters for the Andalusian Centre for Performing Arts from 1992 to 2010, and was again restored to its use as a scenic place in 2016, when it became the headquarters of the Bienal de Flamenco. When used as a scenic space for the Bienal de Flamenco, the public sits on wooden chairs, with the sound source located on a small stage on the main altar, while for theatrical performances the sound source is placed under the dome (Figure 4).

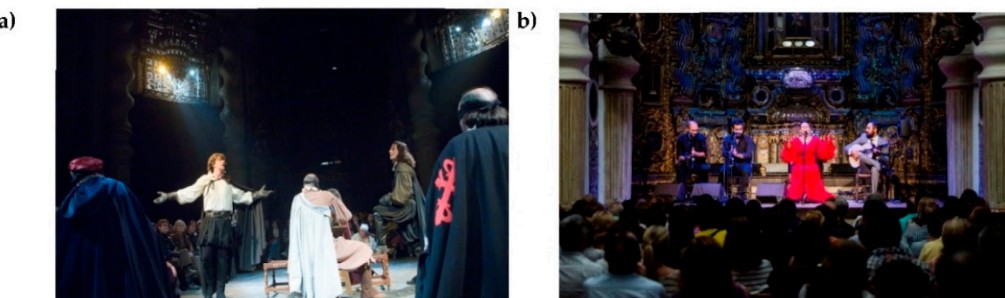

**Figure 4.** (**a**) Representation of the play Don Juan Tenorio. Classical Theatre Company of Seville (2005). http://www.clasicodesevilla.com/Don-Juan-Tenorio. (**b**) María Terremoto. Bienal de Flamenco, Seville (2016). http://www.labienal.com/galeria/.

## 3. Methodology

The characteristics of the sound field in the church of San Luis were studied through in situ measurements of the acoustic conditions of the empty church in its current state and the creation of three-dimensional models reproducing the configurations of sources and audience areas of audience between the 18th and 21st centuries, allowing analysis of the characteristics of the sound field. The CATT-Acoustic software [25] developed by the company CATT (*Computer Aided Theatre Technique*) from Gothenburg (Sweden) was used for this analysis, with two different calculation engines, CATT-*Acoustic v.9.1b* and *CATT TUCT v2.0b*.

For each of the models, the interior of the church has been reproduced using *SketchUp v.14* software, which allows the geometry to be exported to the *CATT* program. The models made simplify the interior space, eliminating any elements which, due to their smaller size, are not decisive when it comes to obtaining adequate results. However, the acoustic properties of the materials of the different surfaces (absorption and scattering) are more decisive for obtaining adequate results [26]. The environmental conditions (humidity and temperature), as well as the background noise obtained from in situ measurements, were considered for the simulations with *TUCT algorithm* 2 (Table 2).

**Table 2.** Calculation conditions.

| Background Noise (dB) | 125 | 250 | 500 | 1000 | 2000 | 4000 |
|---|---|---|---|---|---|---|
| | 43 | 43 | 49 | 52 | 48 | 41 |
| Environmental parameters | Temperature | | | 18.70 °C | | |
| | Humidity | | | 73.00% | | |
| Calculation conditions | Calculation algorithm | | | 2 | | |
| | Number of rays | | | 100,000 | | |
| | Echogram/impulse response | | | 4000 ms | | |
| | Air density | | | 1.20 kg/m$^3$ | | |
| | Air absorption | | | activated | | |
| | Diffraction | | | activated | | |
| Number of planes | Hypothesis 0 (H0) | | | 764 | | |
| | Hypothesis 1 (H1) | | | 797 | | |
| | Hypothesis 2 (H2) | | | 797 | | |
| | Hypothesis 3 (H3) | | | 774 | | |
| | Hypothesis 4 (H4) | | | 774 | | |
| | Hypothesis 5 (H5) | | | 791 | | |
| | Hypothesis 6 (H6) | | | 817 | | |

### 3.1. Acoustic Parameters

For the evaluation of the sound field of San Luis de los Franceses, objective acoustic parameters are obtained from in situ measurement following standard ISO 3382-1 [27].

These parameters are associated with a subjective characteristic that allows the listener to assess the acoustics of the room. After validation of the church model in its current state, the characteristics of the sound field in the different evolutionary models were assessed. The reverberation of the inner space is quantified with the $T_{30}$ parameter, while the perceived reverberation is judged from the Early Decay Time EDT, more closely linked to the subjective reverberation time. The sound strength parameter $G$ is used to assess the subjective sound level; the perceived musical clarity of sound, $C_{80}$; definition, $D_{50}$; the apparent source width from early lateral energy fraction, $J_{LF}$; and listener envelope from the early inter-aural cross correlation coefficient $IACC_E$. The results for all these parameters are obtained spatially averaged by frequency, between 125 and 4000 Hz. The Speech Transmission Index parameter, STI, using a standardized scale to rate the intelligibility of speech based on a standardized scale, is also analysed [28].

### 3.2. In Situ Measurement

Measurements in the church were carried out in situ without an audience to characterize its behaviour [27]. The measurement chain was used to obtain impulse responses (IR) from which all the acoustic parameters that define the sound field can be obtained. At each point where a receiver was located, the IR was obtained from sweeps of sinusoidal wave signals, with frequency increasing exponentially with time [28]. Both the frequency range and the duration of each sweep were adjusted to suit the environmental conditions in order to have impulse responses of adequate quality, so that the signal-to-noise ratio exceeded 45 dB in all the octave bands analysed, 125 to 4000 Hz.

WinMLS2004 software with a Roland Edirol UA 101 sound card was used to generate the signal, recordIRs, and analyse results. The signal generated by the laptop was fed to a Behringer Eurolive B1800DProas amplifier connected to an AVM D012 01 dB omni-directional source. IRs were recorded using an Audio-Technica AT4050 multi-pattern microphone with omnidirectional configuration connected to the Soundfield polarization source (SMP 200). A Head Acoustic HMS III torso simulator pointed towards the sound source was used, together with the OPUS 01dB signal conditioner, to obtain the cross-correlation coefficients. The background noise spectrum is measured with a Brüel & Kjær B&K 4165 microphone connected to a Svantek SVAN 958 noise analyser.

In the acoustic measurement of the church, the sources were placed in four positions. Positions S1 (main altar), S2 (dome) and S3 (choir) corresponded to the positions described in Figure 3, used in the study of the different hypotheses. Source S4, on a side altar, was considered only as an in situ measurement. The positions for the receivers located in the audience area coincided with those in Figure 3, numbered from 1 to 9. The sources and receivers were placed at heights of 1.50 m and 1.20 m, respectively.

With respect to the interior material used (Table 3), the plaster of the walls and dome accounts for more than half of the surfaces, while the marble of the flooring and decorative elements also has a great impact. The wooden altarpieces also occupy a large surface area, with other materials such as glass or ceramic flooring found, albeit to a lesser extent, on the upper floor.

**Table 3.** Materials, location, surface area, and percentage.

| Materials | Location | Surface Area (m$^2$) | Surface Area. (%) |
| --- | --- | --- | --- |
| Plaster | Walls and dome | 1527.80 | 58.62 |
| Marble | Ground and choir floor. Solomonic and choir columns. | 577.34 | 23.15 |
| Wooden altarpieces | Altars | 283.70 | 10.90 |

**Table 3.** *Cont.*

| Materials | Location | Surface Area (m$^2$) | Surface Area. (%) |
|:---:|:---:|:---:|:---:|
| Glass | Dome and choir windows | 82.80 | 3.20 |
| Hole | Communication gap with the Novitiate | 43.70 | 1.70 |
| Ceramic flooring | Choir and tribunes floor. | 38.26 | 1.50 |
| Organ | Choir organ | 23.50 | 0.93 |

Table 4 shows the results for the acoustic parameters of the ISO 3382-1 standard, obtained for the position of the source on the main altar (S1), the source position considered for the validation of the computer model.

**Table 4.** Acoustic parameter values in frequency octave band and single number frequency averaging [27].

| | 125 Hz | 250 Hz | 500 Hz | 1 kHz | 2 kHz | 4 kHz | Single Number |
|:---:|:---:|:---:|:---:|:---:|:---:|:---:|:---:|
| $T_{30}$ (s) | 3.59 | 3.72 | 3.72 | 3.39 | 2.81 | 2.16 | 3.55 |
| EDT (s) | 3.030 | 3.09 | 3.03 | 2.75 | 2.20 | 1.71 | 2.89 |
| G (dB) | 15.07 | 15.63 | 14.28 | 10.97 | 10.37 | 9.42 | 12.62 |
| $C_{80}$ (dB) | −0.42 | −1.18 | −1.80 | −0.61 | 0.41 | 1.52 | −1.20 |
| $D_{50}$ (-) | 0.38 | 0.32 | 0.27 | 0.35 | 0.40 | 0.45 | 0.31 |
| $J_{LF}$ (-) | 0.13 | 0.27 | 0.40 | 0.35 | 0.31 | 0.28 | 0.28 |
| $IACC_E$ (-) | 0.96 | 0.84 | 0.58 | 0.57 | 0.40 | 0.34 | 0.51 |

Based on the measurement results evaluated by Alberdi et al. [20], it can be stated that *"the analysis showed negligible differences in the reverberation time for all sources, as also confirmed by conventional criteria proposed by standards to evaluate the curvature of decays. Nevertheless, for EDT values, the differences in early energy growth gave rise to a different behaviour as a function of the position of the source clearly appeared. Bayesian analysis showed that several double slopes appeared in decay curves, spanning different frequencies, particularly when the source was under the dome or close to a lateral altar. The most affected octave band frequencies were 2 kHz and 4 kHz. Although not particularly evident, the double slope phenomenon could be associated to an uneven distribution of sound energy due to acoustic coupling between different sub-volumes"*. The influence of the large volume of the dome in relation to the perception of the sound field for the different relationships between the position of the source and the receivers is evaluated using the computer models created for the different hypotheses.

*3.3. Model Validation*

Once the data of the model of the current state with the empty church had been entered in CATT-Acoustic, it was validated to adjust it to the results obtained in the in situ measurement, so that its acoustic behaviour was similar to the current one (Figure 5).

To achieve this objective, the absorption and scattering coefficients of the materials, especially those which present greater uncertainty, must be modified. This adjustment was applied to the reverberation time ($T_{30}$) so that the values obtained in the calculation were similar to those measured in situ for the different octave bands. As a validation criterion [29], it was estimated that the coincidence was adequate if they differed less than the perceptible threshold Just Noticeable Difference (JND), that is, 1 JND for $T_{30}$ (less than 5% of the values measured for each octave band). The rest of the parameters were considered adequate below 2 JND, according to consensus [30]. In contrast, as indicated by Martellotta [31] in his study for the value of JND in $C_{80}$ and Ts for three churches, the value of 1 JND for clarity and sharpness in widely reverberant spaces can be modified in relation to the regulations in place, considering 1.5 dB for $C_{80}$. In the case studied, although

a faithful adjustment of the reverberation times was achieved, the validation criterion could not be met in almost any of the parameters. EDT values, corresponding to the early decay time, are more closely related to the subjective perception of reverberation, given its dependence on the energy associated with the early reflections obtained in the in situ measurement. Major variations in EDT were observed based on the position of the sound source and in its behaviour in relation to the source–receiver distance.

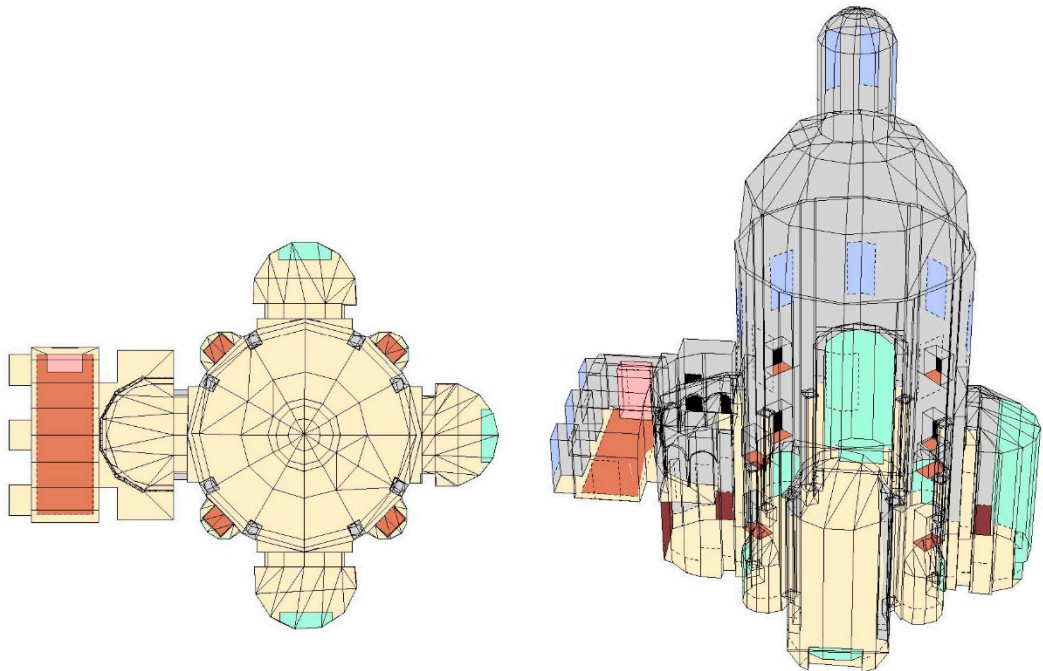

**Figure 5.** San Luis de los Franceses church. Colours correspond to different materials.

This highlights the presence of double slopes in the energy decay curves, showing that the dome behaves as a coupled space. For this reason, EDT was chosen as the adjustment parameter, allowing improved behaviour of most of the acoustic parameters which determine the sound characteristics of the church. The model was validated with the source located at the main altar (S1) and the receivers in the positions used in the in situ measurement. Table 5 shows the absorption and scattering coefficients of the different materials found inside the room used for the validation of the model.

The composition of the plaster walls, which account for more than 50% of the total interior surface, is not known exactly. Therefore, in the interactive process of fitting the model, the initial absorption coefficient values were modified to ensure that the space simulated presents the same acoustic behaviour as the real room.

Another aspect to consider in the model is that the room is not independent from the novitiate building, as the church and novitiate are connected by spaces with no door. These connections occur at the levels of the choir floor and the upper tribune, in the spaces arranged for the visual connection between both parts of the building, considered in the modelling as flat virtual surfaces with an absorption coefficient close to 100%.

Three options were considered for the scattering coefficients. In general, a default value of 10% was allowed in all octave bands for all materials, except for those with an irregular real surface, which were simulated using flat surfaces, as is the case of altarpieces. In this case, variable dispersion coefficients of between 30 and 80% were considered for the octave bands. For the spaces connecting with the rest of the novitiate units, a surface was modelled with an absorption coefficient close to 100% and a dispersion of 1%.

**Table 5.** Finishes: materials, references, surface (%), and absorption (up) and scattering (down) coefficients. * Material used for the adjustment.

| Material | Area (%) | 125 Hz | 250 Hz | 500 Hz | 1 kHz | 2 kHz | 4 kHz |
|---|---|---|---|---|---|---|---|
| Plaster * | 58.60 | 0.12 | 0.12 | 0.11 | 0.13 | 0.15 | 0.17 |
| | | 0.10 | 0.10 | 0.10 | 0.10 | 0.10 | 0.10 |
| Marble [30] | 23.10 | 0.10 | 0.10 | 0.10 | 0.20 | 0.20 | 0.20 |
| | | 0.10 | 0.10 | 0.10 | 0.10 | 0.10 | 0.10 |
| Altarpieces [29] | 10.90 | 0.12 | 0.12 | 0.15 | 0.15 | 0.18 | 0.18 |
| | | 0.30 | 0.40 | 0.50 | 0.60 | 0.70 | 0.80 |
| Wooden door [30] | 0.80 | 0.14 | 0.10 | 0.06 | 0.08 | 0.10 | 0.10 |
| | | 0.10 | 0.10 | 0.10 | 0.10 | 0.10 | 0.10 |
| Glass [32] | 3.2 | 0.04 | 0.04 | 0.03 | 0.03 | 0.02 | 0.02 |
| | | 0.10 | 0.10 | 0.10 | 0.10 | 0.10 | 0.10 |
| Opening | 1.70 | 0.99 | 0.99 | 0.99 | 0.99 | 0.99 | 0.99 |
| | | 0.10 | 0.10 | 0.10 | 0.10 | 0.10 | 0.10 |
| Ceramic flooring [30] | 0.80 | 0.02 | 0.02 | 0.03 | 0.03 | 0.04 | 0.05 |
| | | 0.10 | 0.10 | 0.10 | 0.10 | 0.10 | 0.10 |
| Organ [33] | 0.9 | 0.12 | 0.14 | 0.16 | 0.16 | 0.16 | 0.16 |
| | | 0.30 | 0.40 | 0.50 | 0.60 | 0.70 | 0.80 |

Following the comparison of the results of the in situ measurement and the computer-simulated model, it can be concluded that the model adopted for the church of San Luis de los Franceses adequately reproduces the acoustic behaviour of the room in its current state.

Figure 6 shows the JND differentials of the different parameters, obtained as the difference between the mean values of the parameters at the different frequencies, from the modelling and the in situ measurement, and divided by the JND value obtained in the measurement from the indications given in of UNE-EN ISO 3382-1 standard.

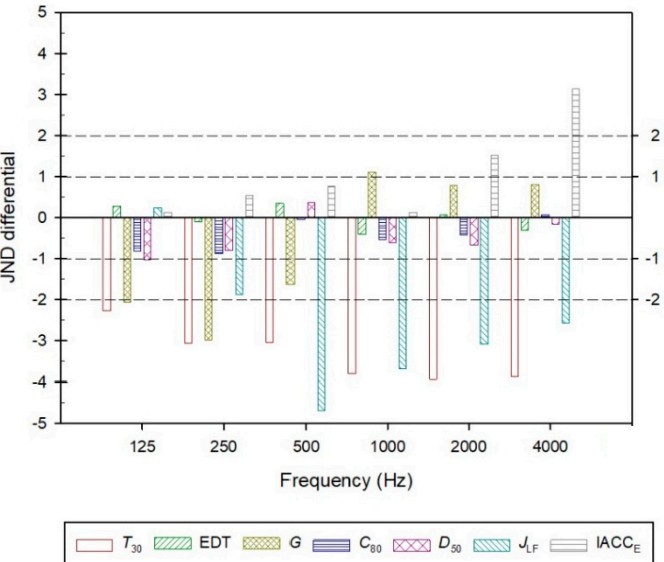

**Figure 6.** JND differentials spatially averaged versus frequency octave band for the acoustic parameters evaluated: difference between the simulated and measured results divided by JND of the measured value.

In general, the results are between −1.00 and +1.00 for the different frequencies. The value of early lateral energy fraction ($J_{LF}$) is that with the lowest adjustment as this parameter is difficult to simulate. However, the $IACC_E$ value is adequate, except for in the 4000 Hz band. The JND differential for reverberation time is higher than the rest of the parameters, obtaining results between −2.00 and −4.00 JND, possibly due to the arrangement of coupled spaces.

*3.4. Analysis of the Evolution of Acoustic Conditions*

After validating the model of the empty church in its current state, the different audience and source positions from the 18th to 21st centuries were studied. Models were drawn up to simulate these changes and any variations in coverings and furniture, allowing the analysis of the characteristics of the sound field in each of the stages studied.

## 4. Evolutionary Models

The necessary modifications are carried out on the initial model to adapt it to the models representing the different evolutionary stages in the history of San Luis, from its establishment in the 18th century to the present day (Figure 7). Six specific hypotheses were established (Table 6) to represent the historical moment (18th-19th century, 20th century, 21st century), the position of the source (S1, S2, S3), and the configuration of the audience (standing, wooden benches, wooden chairs).

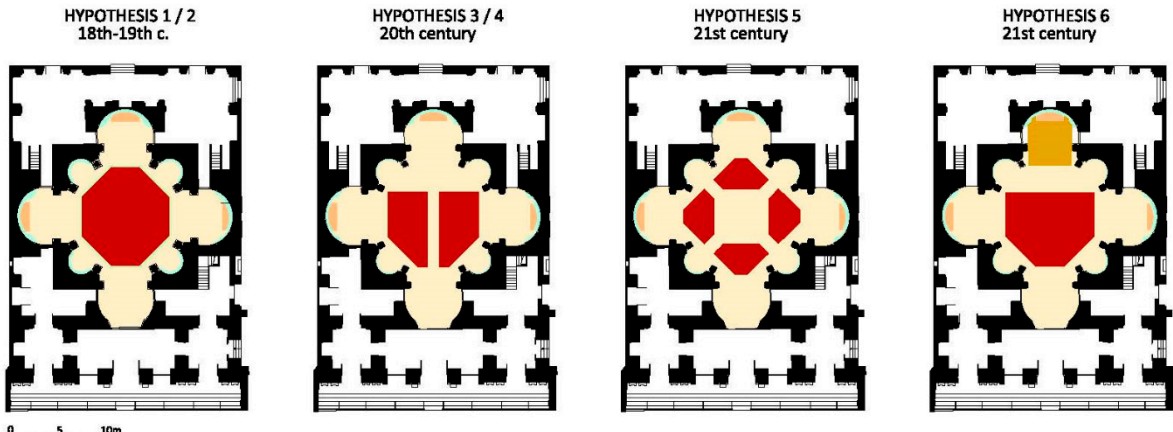

**Figure 7.** Ground floor. Hypothesis audience distribution 1 to 6.

**Table 6.** Evolutionary hypotheses (* liturgy).

|  | H1 * | H2 * | H3 * | H4 * | H5 (Theatre) | H6 (Flamenco B.) |
|---|---|---|---|---|---|---|
| 18–19th c. | X | X |  |  |  |  |
| 20th c. |  |  | X | X |  |  |
| 21st c. |  |  |  |  | X | X |
| S1 (main altar) | X |  | X |  |  | X |
| S2 (dome) |  |  |  |  | X |  |
| S3 (chair) |  | X |  | X |  |  |
| Standing audience | X | X |  |  |  |  |
| Wooden benches audience |  |  | X | X |  |  |
| Wooden seats audience |  |  |  |  | X | X |
| Novices in tribune | X | X |  |  |  |  |
| Stage in main altar |  |  |  |  |  | X |

Table 7 defines the acoustic properties (absorption and scattering) of the modified elements with respect to the initial model, specifically, the planes that simulate the audiences in the different hypotheses, as well as the wooden stage introduced in Hypothesis 6 (H6) and on which S1 is placed 1.50 m above the stage.

**Table 7.** Modified materials, references, and absorption (up) and scattering (down) coefficients.

| Material | 125 Hz | 250 Hz | 500 Hz | 1 kHz | 2 kHz | 4 kHz |
|---|---|---|---|---|---|---|
| Standing audience (1p/m$^2$) [30] | 0.16 0.10 | 0.29 0.10 | 0.55 0.10 | 0.80 0.10 | 0.92 0.10 | 0.90 0.10 |
| Wooden occupied bench [34] | 0.23 0.30 | 0.37 0.40 | 0.83 0.50 | 0.99 0.60 | 0.98 0.70 | 0.98 0.80 |
| Wooden occupied chair (2chairs/m$^2$) [30] | 0.24 0.30 | 0.40 0.40 | 0.78 0.50 | 0.98 0.60 | 0.96 0.70 | 0.87 0.80 |
| Stage: wooden platform [30] | 0.18 0.10 | 0.12 0.10 | 0.10 0.10 | 0.09 0.10 | 0.08 0.10 | 0.07 0.10 |

With respect to audience receivers, those numbered in Figure 2 and within the audience area of each hypothesis are considered. As for the eight positions representing the novices located in tribunes (choir level and upper tribune) of Hypotheses 1 and 2, these correspond to the receivers numbered from 22 to 29, as indicated in Figure 8.

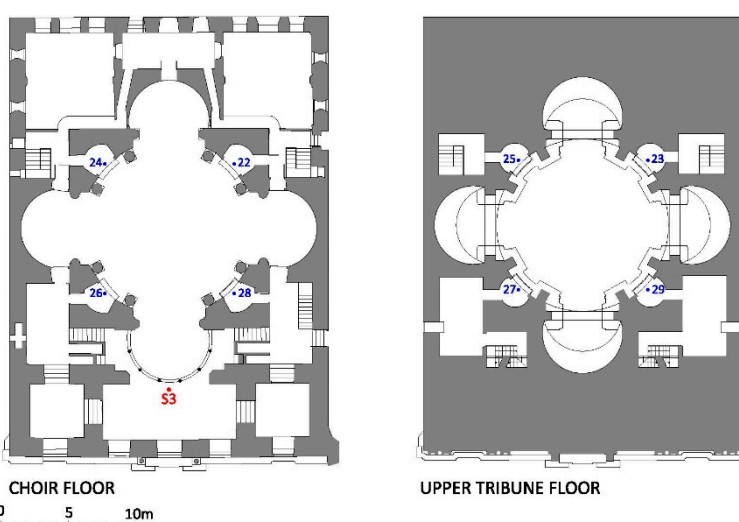

**Figure 8.** Choir and upper tribune floor. Receivers located in stands for novices.

## 5. Results and Discussion

The acoustic behaviour of the church of San Luis de los Franceses was analysed, taking into account the different uses, occupations and sound sources throughout history. Six acoustic simulation models were generated: H1 and H2, corresponding to religious use and with the audience standing in the 18th and 19th centuries; H3 and H4 with the audience seated on wooden benches in the 20th century; H5 and H6 with the church desacralized for the 21st century; H5 for use as a theatre; and H6 for concerts within the Bienal de Flamenco. In H1, H3 and H6 the sound source is located in the main altar, in H2 and H4 in the choir and in H5 in the centre of the audience under the dome.

To evaluate the sound sensation of the listeners in each model, together with the reverberation time ($T_{30}$), the acoustic parameters related to the different subjective sensations are presented: Perceived reverberance (EDT), Subjective level of sound ($G$), Perceived clarity of sound ($C_{80}$ for music and $D_{50}$ for speech), Apparent source width ($J_{LF}$), and Listener envelopment (IACC$_E$).

### 5.1. Global Analysis

Figure 9 represents the spatially averaged values versus frequency in octave bands and the standard deviations of these parameters. Reverberation, evaluated based on the values of $T_{30}$ and EDT, even with the presence of the public, in all models over the centuries

is excessive, both for religious music and for speech. Only the values obtained for the 4000 Hz octave band fall within the optimum range [35]. The averaged values and their errors for $T_{30}$ and EDT are very similar in each octave band, both in time and between both parameters.

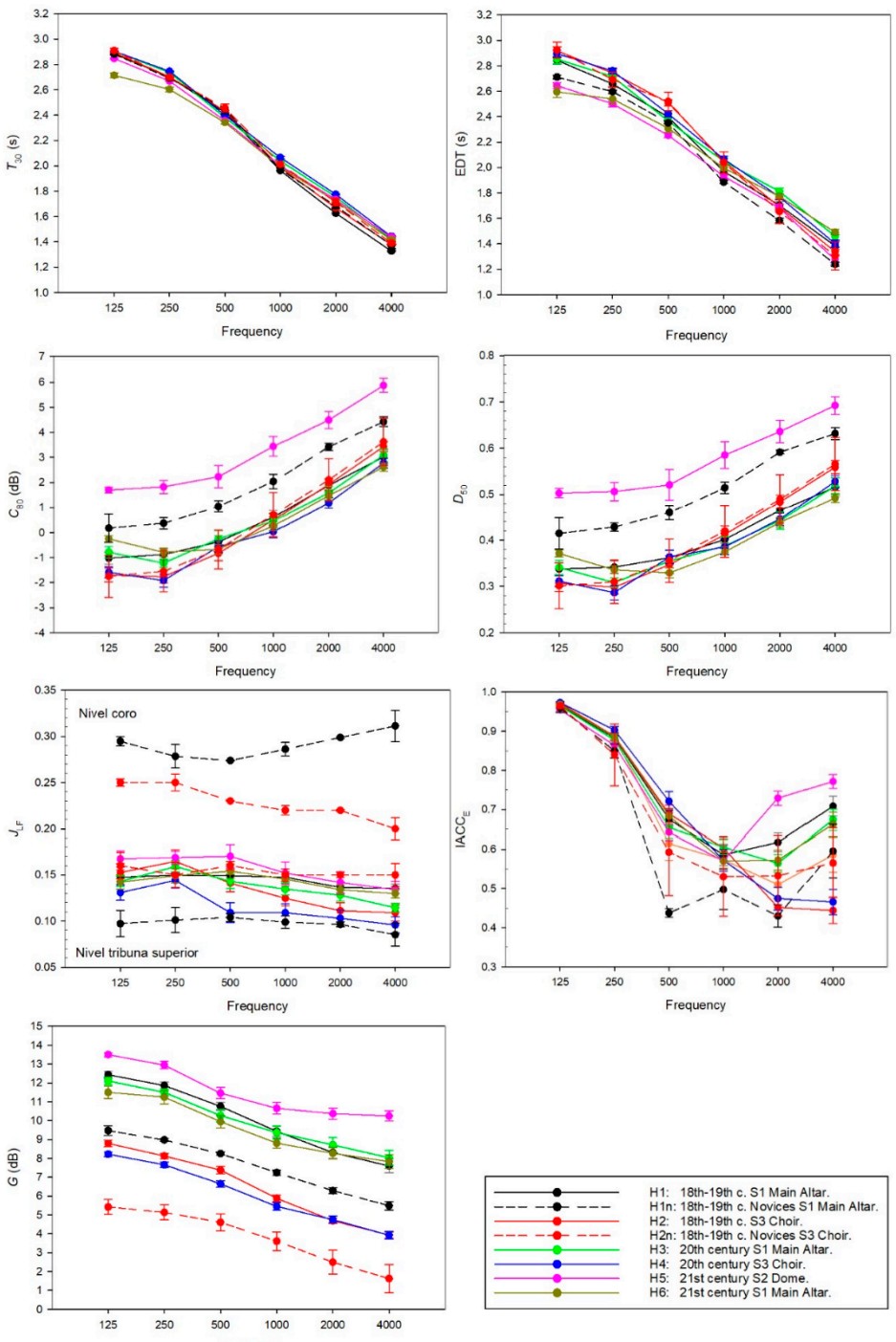

**Figure 9.** $T_{30}$, EDT, $G$, $C_{80}$, $D_{50}$, $J_{LF}$ and IACC$_E$ spatially averaged versus frequency octave band and error bars (standard error) for the different simulation hypotheses.

$C_{80}$ and $D_{50}$ follow the same behaviours and trends. The 21st-century setting for theatre performances displays the best performance. The improvement experienced by the novices in the 18th and 19th centuries when the source is placed on the main altar

is especially noteworthy. In the rest of the models, the behaviour is very similar, with significant differences only at low frequencies (125 Hz).

In addition to the previous models, the figure showing the sensation of the width of the source ($J_{LF}$) incorporates the analysis of the novices in the tribunes at the height of the choir and in the upper floor due to their different behaviour. This is not evident in the rest of the parameters. The differences between the two tribunes are revealed in the first temporary models (H1 and H2), with the source located in the main altar or choir, and with the best subjective sensations being obtained in the upper tribune due to the permanence of the sound late reflections in the dome lantern. Interestingly, when the source is placed in the main altar, the best source width is achieved in the upper tribune and the worst in the lower ones.

For the audience located on the ground floor, the best sensation appears with the source located under the dome, followed by the models that locate the source in the main altar, and the worst results are obtained when the source is located in the choir. In this last case, the subjective sensation worsens in the 20th century compared to the previous ones.

The enveloping sensation of the listener, evaluated in the octave bands of interest (500–2000 Hz) is the most suitable for novices in the 18th-19th centuries, followed by the models where the source is placed in the choir. Next, are the models in which, throughout history, the sound source has been located in the main altar and, lastly, due to the location of the source, the 21st century model for theatrical performances under the dome.

Finally, the *G* values are high enough in all models to create an adequate subjective sound level. The highest values correspond to the 21st-century model, with the source located under the dome. When it is located on the main altar, the sound level remains similar in its temporal evolution to a choir position, although there is a significant decrease compared to the main altar. The same happens to the novices in stands with respect to the faithful on the ground floor.

For the purpose of a global qualification, Table 8 presents the results of the simulations, spectrally and spatially averaged for each of the acoustic parameters, sources and hypotheses. In addition, the receivers located on the ground floor and those located in novices' tribunes in the 18th century are analysed separately, since the results show significant differences in some parameters. The lowest and highest values of the acoustic parameter analysed have been highlighted in red and blue, for the audience on the ground floor and in the stands. The largest and smallest of the audience areas are shaded in the same colours.

**Table 8.** Unique number values for sources S1 (main altar), S2 (dome) and S3 (choir) in the different hypotheses.

| | | $T_{30m}$ (ms) | $EDT_m$ (ms) | $G_m$ (dB) | $C_{80m}$ (dB) | $D_{50m}$ | STI | $J_{LFm}$ | $IACC_{Em}$ |
|---|---|---|---|---|---|---|---|---|---|
| **Ground Floor Audience Receivers** | | | | | | | | | |
| S1 MAIN ALTAR | H1 | 2.19 | 2.18 | 10.10 | 0.12 | 0.38 | 0.51 | 0.15 | 0.63 |
| | H3 | 2.21 | 2.21 | 9.82 | 0.11 | 0.37 | 0.51 | 0.14 | 0.61 |
| | H6 | 2.17 | 2.15 | 9.38 | −0.20 | 0.35 | 0.50 | 0.15 | 0.61 |
| S2 DOME | H5 | 2.18 | 2.09 | 11.06 | 2.83 | 0.55 | 0.56 | 0.16 | 0.65 |
| S3 CHOIR | H2 | 2.21 | 2.28 | 6.63 | −0.16 | 0.38 | 0.52 | 0.15 | 0.58 |
| | H4 | 2.23 | 2.25 | 6.05 | −0.26 | 0.37 | 0.51 | 0.12 | 0.59 |
| **RECEIVERS NOVICES' TRIBUNES** | | | | | | | | | |
| SOURCE | MODEL | $T_{30m}$ (ms) | $EDT_m$ (ms) | $G_m$ (dB) | $C_{80m}$ (dB) | $D_{50m}$ | STI | $J_{LFm}$ | $IACC_{Em}$ |
| S1 MAIN ALTAR | H1 | 2.12 | 2.21 | 7.75 | 1.54 | 0.49 | 0.57 | 0.20 | 0.46 |
| S3 CHOIR | H2 | 2.21 | 2.27 | 4.11 | 0.02 | 0.39 | 0.52 | 0.20 | 0.55 |

## 5.2. Evolutionary Analysis

The values for reverberation time, calculated using the parameter $T_{30m}$, are very similar throughout history for the different uses. In general, the variation between the different models in reverberation times is within a range of 4.00%, regardless of the position

of the source. This was foreseeable, since there are no significant changes in volume or in the absorption of the coatings of the church in the models (Figure 10). In addition, if these average values are compared with the average values of the optimal reverberation times, the church shows a degree of reverberance since in all cases they remain above the recommended limit, 1.59 s for musical use and 1.19 s for speech [35].

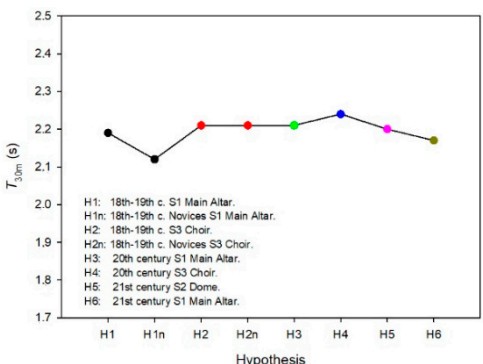
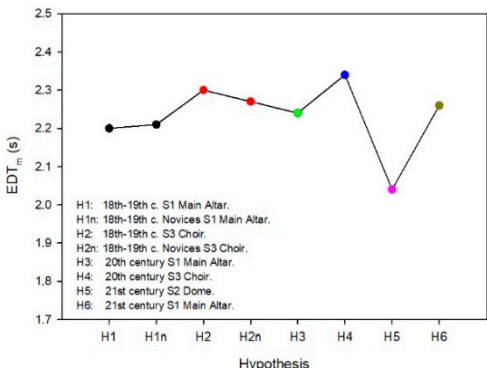

**Figure 10.** Acoustic parameter evolution $T_{30}$ and EDT. Values averaged to a single number.

EDT results support those obtained for $T_{30m}$. The lowest reverberation occurs in the 21st-century layout when the source is located under the dome that exceeds 1 JND compared to the hypotheses of the 18th to 20th centuries.

The degree of amplification produced by the audience room located on the ground floor or in the stands is lower when the source is located in the choir when compared to its location on the main altar or under the dome, exceeding 2.5 JND. In addition, the subjective sound level of the novices is lower than that of the audience on the ground floor.

Figure 11 represents the distribution of *G*, in the ground floor audience area, at a frequency of 1000 Hz, for the different hypotheses. No major differences are observed in the behaviour of the room between hypotheses H1–H2 and H3–H4, which correspond to the 18th–19th centuries and the 20th century, respectively. However, the introduction of wooden benches for the liturgy in the 20th century leads to a decrease of 1 JND in the perception of the sound level for the audience located in the benches when the sound source is located in the main altar. This last hypothesis, H3, is very similar to that obtained in the 21st century for the Bienal de Flamenco (H6). In H5, when the sound source is placed under the dome, the amplification of the room increases in the audience area on the ground floor due to the decrease in distances to receivers, and a very similar spatial distribution is achieved for the four audience distributions.

The sound clarity perceived, both musical, evaluated using $C_{80m}$, and speech, through $D_{50m}$, presents values that remain below 1 JND over time for the different sound sources and audiences. However, two exceptions must be noted. The clarity of the novices during the 18th and 19th centuries is clearly superior to that of the audience on the ground floor when the source is located in the main altar. In addition, the arrangement of the audience in the 21st century, with the source under the dome, obtains the best results of musical clarity.

If evaluating the intelligibility of the church using the STI (Speech Transmission Index) parameter, the results confirm the results of $C_{80m}$ and $D_{50m}$, allowing the results to be qualified within the acceptable range (0.45–0.60).

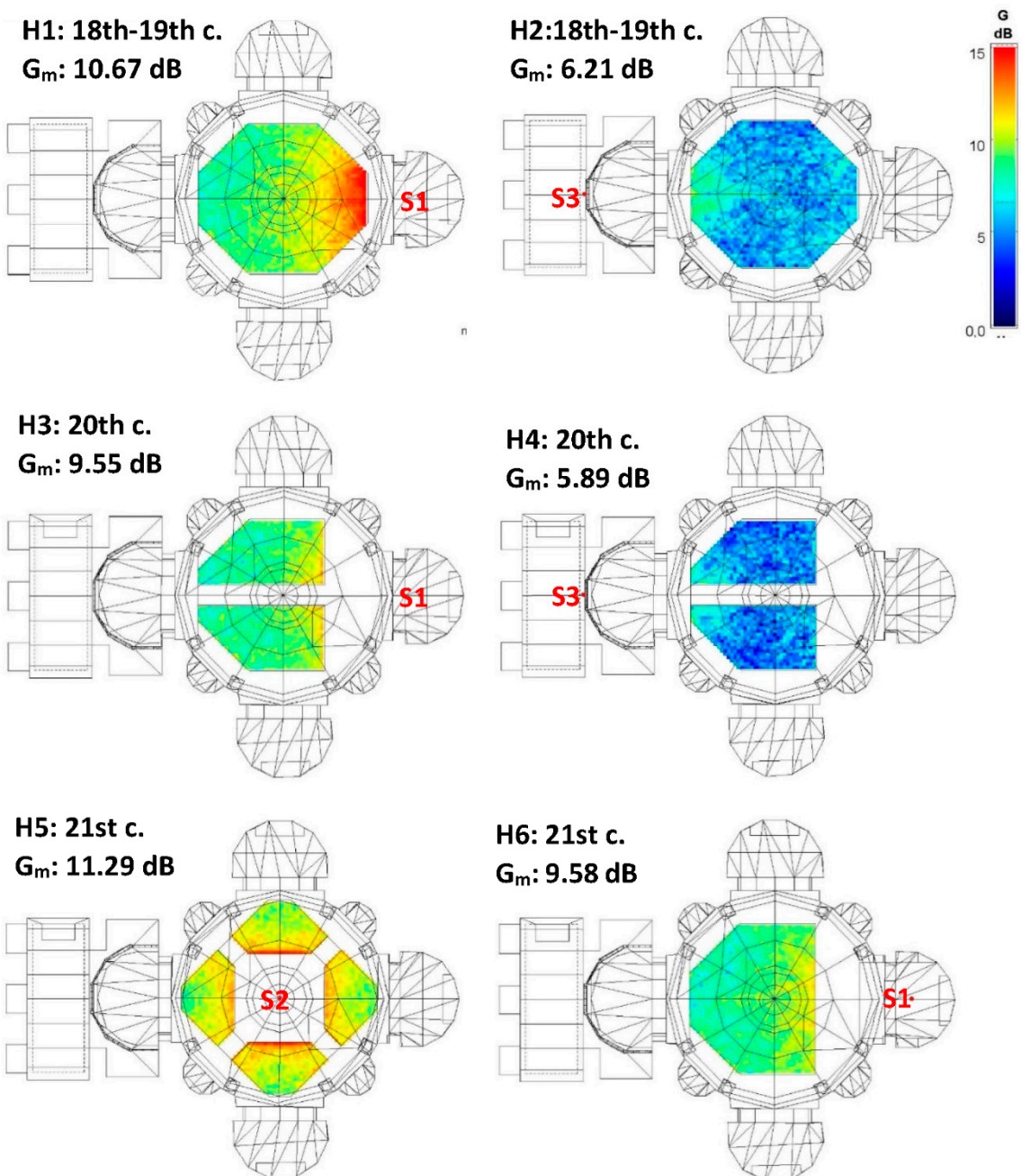

**Figure 11.** Sound force mapping 1 kHz octave band and for the different hypotheses. S1 (main altar), S2 (dome) and, S3 (choir).

When evaluating the apparent sensation of the source, the $J_{LFm}$ values obtained for the ground floor audience for all models are low (<0.20), with variations between positions and hypotheses below 1 JND. However, for the novices' positions analysed in the 18th and 19th centuries, the values are the highest, so that the differences with respect to the public on the ground floor are 1 JND.

The analysis is concluded with an evaluation of the enveloping sensation of the listener using the parameter $IACC_{Em}$. For the receivers of the ground floor the sensation of spatiality has remained constant over time, with a range of values less than 1 JND and showing the best results when the source is located in the choir. There is a notable improvement for the novices with respect to the audience on the ground floor, especially when the source is located in the main altar.

A global evaluation of the results shows that, for the audience on the ground floor, when the source is located under the dome there is an improvement in all subjective

sensations, while similar results to the rest of hypotheses are obtained for the enveloping sensation. The results for receivers located on the ground floor and the source in the main altar (S1) are worse that those with the source located under the dome (S2). The best results correspond to the H1 model, which simulates the Catholic liturgy in the 18th-19th centuries with the parishioners standing. Notably, the worst musical clarity results are found in the current arrangement for the Bienal de Flamenco (H6). When the source is located in the choir (S3), the acoustic conditions are the same as when the source is located in the main altar (S1), with a worsening in reverberation, sound level and musical clarity. The acoustic conditions of the listeners on the ground floor are very similar, although with a slight worsening in the 20th century compared to the two preceding centuries.

Furthermore, the novices' tribunes, during the first two centuries of the study, present better subjective acoustic sensations when the sound source is placed in the main altar. Compared to the rest of the hypotheses, there is an improvement in perceived clarity and a better spatial sensation in the tribunes.

*5.3. Strengths and Limitations*

The ISO 3382-1 standard is applicable in performance spaces. The deconsecrated church of San Luis de los Franceses is currently being used to stage theatrical performances, concerts and cultural events and therefore the application of this standard would be relevant. However, it must be emphasized that this standard was designed for concert halls and auditoriums whose acoustic behaviour differs greatly from that of churches, where absorption distribution is not homogeneous and the presence of chapels, domes and aisles results in a distribution of energy far from the diffuse field. In this case, depending on the position of the source and the receiver, as well as the cupola or choir which act as coupled spaces, the decrease in energy can cease to be linear, giving rise to two or more slopes. Caution must therefore be exercised when considering reverberation time. For the same reason, early energy is highly variable and intrinsic to the location of the source and receiver, so that EDT loses the homogeneity of values obtained in concert halls and auditoriums. The main limitation of the application of the standard consists of the JND associated with each acoustic parameter established in theatres. The Martellota [31] criteria for $C_{80}$ and Ts in this type of space have been taken into account for the analysis carried out. Finally, the typical values collected in the standard for each acoustic parameter do not have to coincide with those obtained in churches.

Moreover, acoustic simulations are also subject to limitations. The main limitation is connected with the wave nature of the sound "rays" not being taken into account. In this case, the Schroeder frequency of each evolutionary period is around 45 Hz and therefore statistical acoustics would be valid from 250 Hz. Although all the simulations estimated the effect of diffraction and scattering, they were not able to capture the real wave phenomena. The level of detail used was estimated following the recommendations of Vorländer [30]. However, the treatment of the dome as a set of flat surfaces hinders the adequate reproduction of its focal behaviour. In all hypotheses, the final number of rays assumed was considered so that the results obtained in different simulations of the same hypothesis do not differ by more than 2%. All the absorption and scattering coefficients were taken from the specialized literature. Only retouching below 1% was assumed in plaster to adjust the measured and simulated results.

## 6. Conclusions

The church of San Luis de los Franceses has maintained its spatial configuration in terms of volume and interior cladding since the 18th century, except for the variations resulting from modifications in the furniture for the audience. In the 18th–19th centuries, the public remained standing, whereas in the 20th and 21st centuries, they went on to sit on wooden benches and chairs, respectively. In the case of the performances for the Bienal de Flamenco, a small wooden stage is also set up at the main altar. The plaster covering the walls and domes of the church accounts for over 50% of the surfaces in all models and

presents slight variations which remain within 2% with respect to the total surfaces for all models between the 18th and 21st centuries. The rest of the materials, such as altarpieces, marble or glass, show variations below 1% between models.

Given the continuity in volume and coatings since its inauguration in 1731 until the 21st century, the characteristics of the sound field of the church of San Luis have remained stable and the variations detected are a consequence of the different relationships between the position of the source and the receivers, introduced by different users in different historical periods.

The evolutionary analysis of the acoustic conditions of the church of San Luis de los Franceses makes it possible to state that the model currently used for theatrical performances (20th c.), with the audience distributed around the sound source located under the dome, improves the sound clarity, the perceived source width and the sound strength in the audience located on the ground floor. However, the worst results are obtained when evaluating the listener envelopment.

During the 18th and 19th centuries, the sound clarity and the spatial sensation improved for the novices who attended religious acts in front of the standing audience on the ground floor when the sound source is in the main altar.

**Author Contributions:** Conceptualization, E.A., M.G. and Á.L.L.-R.; methodology, E.A., M.G. and Á.L.L.-R.; software, E.A., M.G. and Á.L.L.-R.; validation, E.A., M.G. and Á.L.L.-R.; formal analysis, E.A., M.G. and Á.L.L.-R.; investigation, E.A., M.G. and Á.L.L.-R.; resources, E.A., M.G. and Á.L.L.-R.; data curation, E.A., M.G. and Á.L.L.-R.; writing—original draft preparation, E.A., M.G. and Á.L.L.-R.; writing—review and editing, E.A., M.G. and Á.L.L.-R.; visualization, E.A., M.G. and Á.L.L.-R.; supervision, E.A., M.G. and Á.L.L.-R.; project administration, E.A., M.G. and Á.L.L.-R.; funding acquisition, E.A., M.G. and Á.L.L.-R. All authors have read and agreed to the published version of the manuscript.

**Funding:** This research received no external funding.

**Institutional Review Board Statement:** Not applicable.

**Informed Consent Statement:** Not applicable.

**Data Availability Statement:** Not applicable.

**Acknowledgments:** The authors wish to thank the Provincial Council of Seville for facilitating access to the church to carry out acoustic measurements and Fernando Mendoza Castells, architect in charge of the restoration of the church, for the graphic documentation provided.

**Conflicts of Interest:** The authors declare no conflict of interest.

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
