# Peer review of "Evolutionary Analysis of the Acoustics of the Baroque Church of San Luis de los Franceses (Seville)"

_applsci, doi:10.3390/app11041402_

Round 1
Reviewer 1 Report
General
Interesting measurement and evaluation work concerning a remarkable historical ecclesiastical space whose use over time has evolved into today's profane use as a space for performances.
The authors investigate this evolution on a room acoustic level, using existing simulation software that is calibrated using measurements.
In this way, the historical evolution of the characteristic room acoustic parameters is investigated.
More detailed
Excellent introduction with clear, sufficiently concise historical context, good positioning in relation to related acoustic research, clear objectives.
Excellent and well-documented account of all aspects of the work done.
Well written text, good figures and tables, relevant analysis.
Minor remarks and suggestions
+ Minor typing errors: 178, Table 2. 73-00 % 1-20 kg/m3
+ 151 diffusion, 222 scattering, and 235 dispersion, all for the same phenomenon? Better to fine-tune this.
+ 182 'Soundfield MKV microphone was used to obtain the directional distribution of each reflection.' The results of this technique do not appear in the submission, please explain or remove.
+ For the sake of completeness and as also required by the ISO Standard, it is necessary to mention how the direction sensitive quantities were obtained. 'Selection and usage of a dummy head shall be clearly stated in the test report, and the direction of the dummy head shall be described in detail.' Cit ISO 3382-1:2009 (E).
If the authors can still update these small points, this reviewer wants to sincerely support the acceptance of this text!
Reviewer 2 Report
The paper is interesting, it deals with an increasingly popular topic in room acoustics that is related to historical acoustics or archaeoacoustics. I think some references could be done to this emerging research field as it tries to position itself in the broader panorama of architectural acoustics. See for instance: (Scarre & Lawson, 2006) (Aletta & Kang, 2020).
The title is a bit too generic and vague: I would recommend something more specific/descriptive and self-explanatory.
The literature review about church acoustics should also be expanded (or other literature reviews should be acknowledged?); see for instance: (Giron et al., 2017) (Suárez, Sendra, Navarro, & León, 2005) (Suárez, Alonso, & Sendra, 2015) (Đorđević, Novković, & Andrić, 2019) (Elicio & Martellotta, 2015) (Alonso, Suárez, & Sendra, 2019).
The methodological part looks rigorous to me and results are clearly presented.
Since D50 for speech is presented and C80 for music, would it make sense to report also C50 for speech? A small addition in the results section should be feasible.
In the Discussions, there are two aspects that should be considered, and probably expanded in a dedicated “Limitations” section:
- Applicability limits of the ISO 3382-1: this is for performance spaces, no standard exists for churches specifically so Part 1 is the closest match. Please elaborate on limits of this standard applicability to the church context
- Uncertainty of the software simulation – this is not discussed at all, and readers should be made aware of all the systematic uncertainties that are likely to propagate to the results. See for instance: (Vorlander, 2013)
References suggested for the next draft:
Aletta, F., & Kang, J. (2020). Historical Acoustics: Relationships between People and Sound over Time. Acoustics, 2(1), 128-130.
Alonso, A., Suárez, R., & Sendra, J. J. (2019). The Acoustics of the Choir in Spanish Cathedrals. Acoustics, 1(1), 35-46.
Đorđević, Z., Novković, D., & Andrić, U. (2019). Archaeoacoustic Examination of Lazarica Church. Acoustics, 1(2), 423-438.
Elicio, L., & Martellotta, F. (2015). Acoustics as a cultural heritage: The case of Orthodox churches and of the “Russian church” in Bari. Journal of Cultural Heritage. doi:10.1016/j.culher.2015.02.001
Scarre, C., & Lawson, G. (2006). Archaeoacoustics. Mc Donald Institute for Archaeological Reserach, University of Cambridge.
Suárez, R., Alonso, A., & Sendra, J. J. (2015). Intangible cultural heritage: The sound of the Romanesque cathedral of Santiago de Compostela. Journal of Cultural Heritage, 15, 239-243.
Suárez, R., Sendra, J. J., Navarro, J., & León, A. L. (2005). The sound of the cathedral-mosque of Córdoba. Journal of Cultural Heritage , 6, 307-312.
Vorlander, M. (2013). Computer simulations in room acoustics: concepts and uncertainties. J. Acoust. Soc. Am.(133).
Giron, S., Alvarez-Morales, L., & Zamarreno, T. (2017). Church acoustics: A state-of-the-art review after several decades of research. Journal of Sound and Vibration, 411, 378-408.
Author Response
Please see the attachment
REVIEWER 2:
- The paper is interesting, it deals with an increasingly popular topic in room acoustics that is related to historical acoustics or archaeoacoustics. I think some references could be done to this emerging research field as it tries to position itself in the broader panorama of architectural acoustics. See for instance: (Scarre & Lawson, 2006) (Aletta & Kang, 2020).
The authors would like to thank the reviewer for this recommendation. This has now been included in the introduction, in lines 62 to 69.
- The title is a bit too generic and vague: I would recommend something more specific/descriptive and self-explanatory
The title of the article has been modified according to the indications made, the new title proposed is:
Evolutionary analysis of the acoustics of the Baroque church of San Luis de los Franceses (Seville)
- The literature review about church acoustics should also be expanded (or other literature reviews should be acknowledged?); see for instance: (Giron et al., 2017) (Suárez, Sendra, Navarro, & León, 2005) (Suárez, Alonso, & Sendra, 2015) (Đorđević, Novković, & Andrić, 2019) (Elicio & Martellotta, 2015) (Alonso, Suárez, & Sendra, 2019).
As requested, the bibliography has been reviewed and works relating to the aspects examined in the research have been included. These have been included in the paragraph in lines 62-69 and line 78. They are included in detail in the references. The inclusion of new references has led to the numbering of all the references of the article being revised. The new references are listed below. Reference 20 from the initial text has been deleted.
References
- Borromei, C. Instructionum Fabricae et Supellectilis Ecclesiasticae. Da Ponte, Pacífico, Milán, Italia, 1557.
- Sendra, J.J.; Navarro, J. La evolución de las condiciones acústicas en las iglesias del paleocristiano al tardobarroco. Eds. Universidad de Sevilla, Instituto Universitario de Ciencias de la Construcción, 1997, pp.52-62.
- Fernando García Gutierrez, S.J. Introducción: Coordenadas Histórico-Geográficas de la Provincia Bética de la Compañía de Jesús (Soto Artuñedo, W.). El arte de la Compañía de Jesús en Andalucía (1554-2004). Publicaciones Obra Social y Cultural. CajaSur, Córdoba, España, 2004, pp. 18-25.
- Rodriguez G. de Ceballos, A. Bartolomé de Bustamante y los orígenes de la arquitectura jesuítica en España. Institutum Historicum S.I., Roma, Italia, 1967, pp. 332.
- Aleta, F., Kang, J. Historical Acoustic: Relationships between People and Sound over Time. Acoustics, 2020 2 (1), 128-130. doi: 10.3390/acoustics2010009
- Scarre, C., Lawson, G. Archaeoacoustics. MacDonald Institute for Archaeological Research. University of Cambridge, 2006.
- Đorđević, Z., Novković, D., Andrić, U. Archaeoacoustic. Examination of Lazarica Church. Acoustic 2019, 1(2), p.423-438. doi: 10.3390/acoustics1020024
- Suárez, R. Alonso, A. Sendra, J.J. Intangible cultural heritage: The sound of the Romanesque cathedral of Santiago de Compostela. Journal of Cultural Heritage 2015, 16, p. 239-243. doi: 10.1016/j.culher.2014.05.008
- Suárez, R., Alonso, A., Sendra, J.J. Archeoacoustics of intangible cultural heritage: The sound of the Maior Ecclesia of Cluny. Journal of Cultural Heritage 2016, 19, p. 567-572. doi: 10.1016/j.culher.2015.12.003
- Alonso, A., Suárez, R., Sendra, J.J. The Acoustic of the Choir in Spanish Cathedrals. Acoustic, 2019, 1, 35-46. doi: 10.3390/acoustics1010004
- Alberdi, E., Galindo, M., León-Rodríguez, A.L. Acoustic behaviour of polychoirs in the Baroque church of Santa María Magdalena, Seville. Applied Acoustics 2021, doi: 10.1016/j.apacoust.2020.107814
- Suárez, R., Sendra, J.J., Navarro, J., León, A.L. The sound of the Cathedral-Mosque of Cordoba. Journal of Cultural Heritage 2005, 6(4) p. 307-312. doi: 10.1016/j.culher.2005.03.05
- Cirillo, E., Martellotta, F. Worship, Acoustic, and Architecture. Multi-Science publishing CO. LTD., Brentwood, Essex, England, 2006, pp. 147-171
- Carvalho, A. Acoustical Measures in Churches Portos´s Clérigos Church. A comprehensive example. Procedings of the 7th ICSV, 4-7 julio 2000, Garmisch-Partenkirchen, Alemania, vol III, p. 1645-1652.
- Tzekalkis, E. G. Reverberation time of the Rotunda of Thesalonikki. Soc. Am. 1975, 57(5), p 1207-1209. doi: 10.1121/1.380545
- Su-Guhl, Z., Yilmazer, S. The Acoustical Characteristics of the Kocate Mosque in Ankara, Turkey, Sci. Rev. 1964, 51, p 21-30. doi:10.3763/asre.2008.5104
- Elicio, L., Martellotta, F. Acoustics as a cultural heritage: The case of Orthodox churches and of the “Russian church” in Bari. Journal of Cultural Heritage 2015, 16, p. 912-917. doi: 10.1016/j.culher.2015.02.001
- Moreno, A., Zaragoza, J. Alcantarilla, F. Generation and suppression of flutter echoes in spherical domes. Acoust. Soc. Jnp. 1981, E2, p.197-202. doi: 10.1250/ast.2.197
- Vercammen, M.L.S. Sound concentration caused by curved surfaces, Technische Universiteit Eindhoven 2012, PhD dissertation. doi: 10.6100/IR732486
- Alberdi, E., Martellotta, F., Galindo, M., León, A.L. Dome sound effects in San Luis. Applied Acoustics 2019, 156, p.56-65. doi: 10.106/j.apacoust.2019.06.030
- De la Banda y Vargas, A. La Iglesia de San Luis de los Franceses. Exma. Diputación Provincial de Sevilla, Sevilla, España, 1977, pp. 19-37.
- Bonet Correa, A. Andalucía Barroca: arquitectura y urbanismo. Barcelona Ediciones Polígrafa, Barcelona, España 1978, pp. 88-91.
- Castilla, M. Influencia del humanismo en la arquitectura de los Jesuitas: Iglesia de San Luis de los Franceses de Sevilla. Liño 23. Revista Anual de Historia del Arte, 2017, p. 21-29. doi: 10.17811/li.2017.21-29
- Pozzo, A. Perspective in architecture and painting. (1642) Eds. New York: Dover, 1989.
- Dalenbäck B.-I. L. 2011. CATT-Acoustic v9 Powered by TUCT. User’s Manual. Gothenburg: CATT
- Rindel, J. H., Shiokawa, H., Christensen, C. L., Gade, A. C. Comparisons between computer simulations of room acoustical parameters and those measured in concert halls. Joint meeting of the Acoustical Society of America and the European Acoustics Association, 1999, Berlín, Alemania, p. AAa3. doi: 10.1121/1.425555
- ISO 3382-1:2009(E). Acoustics-Measurement of room acoustic parameters-Part 1: Performance spaces. International Organisation for Standardisation, Geneva, Switzerland, 2009.
- CEI 60268-16:2011. Equipos para sistemas electroacústicos. Parte 16: Evaluación objetiva de la inteligibilidad del habla mediante el índice de transmisión del habla.
- Galindo, M., Zamarreño18, T. y Girón, S. Acoustic simulations of mudejar-gothic churches. Journal of the Acoustical Society of America, 2009, 126 (3), p. 1207-1218. doi: 10.1121/1.3180632
- Vorländer, M. Auralization, fundamentals of acoustics, modelling, simulation, algorithms and acoustic virtual reality. Eds. Berlin: Springer-Verlag, 2008, pp. 303-310.
- Martellotta, F. The just noticeable difference of center time and clarity index in large reverberant spaces. Journal of the Acoustical Society of America, 2010, 128, p. 654-663. doi: 10.1121/1.3455837
- Arau, H. ABC de la acústica arquitectónica. CEAC Barcelona, 1999, pp. 264.
- Carmona, C., Zamarreño, T., Girón, S. and Galindo, M. Acústica virtual de la iglesia de San Lorenzo de Sevilla, Revista de Acústica 40, 2009, 40, 7-12.
- Martellotta, F., D´alba, M., Della Crociata, S. Laboratory measurement of sound absorption of occupied pews and standing audiences. Applied Acoustics, 2011, 72, p. 341-34. doi: 10.1016/j.apacoust.2010.12.008
- Beranek, L. Eds. American Institute of Physics, Acoustical Society of America, New York, Usa, 1993.
- The methodological part looks rigorous to me and results are clearly presented.
- Since D50 for speech is presented and C80 for music, would it make sense to report also C50 for speech? A small addition in the results section should be feasible.
The C50 graph has not been included in the article because it displays no significant differences with respect to the C80 results for the final research conclusions.
|
C50, D50 and C80 spatially averaged versus frequency octave band and error bars (standard error) for the different simulation hypotheses. |
- In the Discussions, there are two aspects that should be considered, and probably expanded in a dedicated “Limitations” section:
- Applicability limits of the ISO 3382-1: this is for performance spaces, no standard exists for churches specifically so Part 1 is the closest match. Please elaborate on limits of this standard applicability to the church context
- Uncertainty of the software simulation – this is not discussed at all, and readers should be made aware of all the systematic uncertainties that are likely to propagate to the results. See for instance: (Vorlander, 2013)
In order to clarify the aforementioned aspects, some aspects relating to the methodology have been expanded, specifically in section 3.3 (Model validation) described below. These appear in the paragraph of lines 223-237.
Paragraph between lines 264-272. Clarifications on figure 6.
Finally, point 5 (Results and Discussion) has been subdivided into sections. Section 5.1 (Global analysis), from line 323 to line 368, 5.2 (Evolutionary analysis), from line 369 to line 431, and section 5.3, strengths and limitations (line 433 to 459), which responds to the issues raised.

Round 2
Reviewer 2 Report
Thank you for addressing my comments, the manuscript looks improved in my opinion. Just check once again for the bibliography, I have spotted a couple of typos: ref.5 Aleta, should be Aletta; ref 35. (Beranek) is currently not numbered in the pdf version, I think something wrong happened with the formatting.
Nice work!
